# Protocol for a 1-year prospective, longitudinal cohort study of patients undergoing Roux-en-Y gastric bypass and sleeve gastrectomy: the BARI-LIFESTYLE observational study

Friedrich C Jassil,[1,2] Alisia Carnemolla,[1,3] Helen Kingett,[2,3] Bruce Paton,[4] Aidan G O'Keeffe,[5] Jacqueline Doyle,[2,3] Stephen Morris,[6] Neville Lewis,[7] Amy Kirk,[2,3] Andrea Pucci,[1,2] Kusuma Chaiyasoot,[1,2] Rachel L Batterham[1,2,3]

For numbered affiliations see end of article.

**Correspondence to**
Professor Rachel L Batterham;
r.batterham@ucl.ac.uk

## ABSTRACT

**Introduction** Roux-en-Y gastric bypass and sleeve gastrectomy are the two most common bariatric surgery performed in the UK that result in comparable weight loss and remission of obesity-associated comorbidities. However, there is a paucity of studies examining the impact of these procedures on body composition, physical activity levels, sedentary behaviour, physical function and strength, dietary intake, health-related quality of life and costs.

**Methods and analysis** The BARI-LIFESTYLE observational study is a 1-year prospective, longitudinal cohort study within a real-world routine clinical care setting aiming to recruit 100 patients with severe obesity undergoing either primary Roux-en-Y gastric bypass or sleeve gastrectomy from two bariatric centres in London, UK. Participants will be followed up four times during the study period; presurgery baseline (T0) and at 3 (T1), 6 (T2) and 12 months (T3) postsurgery. In addition to the standard follow-up investigations, assessments including dual-energy X-ray absorptiometry scan, bioelectric impedance analysis, 6 min walk test, sit-to-stand test and handgrip test will be undertaken together with completion of questionnaires. Physical activity levels and sedentary behaviour will be assessed using accelerometer, and dietary intake will be recorded using a 3-day food diary. Outcome measures will include body weight, body fat mass, lean muscle mass, bone mineral density, physical activity levels, sedentary behaviour, physical function and strength, dietary intake, health-related quality of life, remission of comorbidities, healthcare resource utilisation and costs.

**Ethics and dissemination** This study has been reviewed and given a favourable ethical opinion by London-Dulwich Research Ethics Committee (17/LO/0950). The results will be presented to stakeholder groups locally, nationally and internationally and published in peer-reviewed medical journals. The layperson summary of the findings will be published on the Centre for Obesity Research, University College London website (http://www.ucl.ac.uk/obesity).

### Strengths and limitations of this study

► A comprehensive prospective, longitudinal study with detailed assessments undertaken prior to and for 1 year following bariatric surgery examining changes in body composition, physical activity (PA) levels, sedentary behaviour, physical function and strength, dietary intake, health-related quality of life and costs, relative to baseline presurgery.

► The use of validated research tools (accelerometer to assess PA levels and sedentary behaviour, dual-energy X-ray absorptiometry (DXA) scan to assess body composition and validated questionnaires) will generate high-quality data.

► The study design does not include a conventional intensive lifestyle intervention (non-surgical) as a comparator group and patients will not be randomised to Roux-en-Y gastric bypass (RYGB) or sleeve gastrectomy (SG) in order to reflect current real-world clinical care.

► A potential sample selection bias due to exclusion of patients with functional limitation and/or non-ambulatory and patients with more than 200 kg of body weight owing to the weight limit of the DXA scan.

► A relatively small sample size, nevertheless, this number is adequate to generate indepth insights into the various outcomes of RYGB and SG as delivered in the UK healthcare setting.

## INTRODUCTION

Bariatric surgery engenders marked sustained weight loss and is recommended by the National Institute for Health and Care Excellence (NICE) as a treatment option for people of severe obesity,[1] estimated to affect approximately 2.6 million adults in the UK.[2] Roux-en-Y gastric bypass (RYGB) and sleeve gastrectomy (SG) are now the two most common procedures performed in the UK, which result in comparable weight loss and

remission of obesity-associated comorbidities.[3] However, there is a paucity of studies examining the impact of these procedures on body composition, particularly bone mineral density (BMD), physical activity (PA) levels, sedentary behaviour, physical function and strength, dietary intake, health-related quality of life (HRQoL) and costs. Furthermore, current eligibility and success criteria of bariatric surgery are mainly based on body weight, body mass index and excess weight loss but evidence have shown various beneficial outcomes of the surgery above and beyond weight loss alone, hence highlighting the need for more functional preoperative and postoperative patient assessment.[4 5]

Bariatric surgery leads to a marked decrease in fat mass (FM), but fat free mass (FFM) particularly bone mass is also reduced postsurgery,[6] potentially negatively impacting on physical function and strength, and putting patients at increased risk of osteoporotic fracture in the future.[7 8] Moreover, a recent study has revealed a positive association between changes in adiposity with cardiometabolic outcomes postsurgery, indicating the usefulness of incorporating body composition assessment.[9] Surgical modification of the gastrointestinal tract impairs the intake and/or absorption of essential nutrients for bone health that consequently perturbs bone metabolism, leading to BMD deterioration.[7 8 10 11] Significant bone mass loss has been reported to occur rapidly in the first year of surgery and continues to deteriorate up to 3 years even after maximum weight loss has been achieved.[10] However, these data are mainly based on studies undertaken in patients who underwent RYGB whereas SG is now the most common procedure undertaken both in the UK and globally.[3 12] Currently, it is unclear whether the rate of bone mass loss following SG parallels weight loss.[13–15] Given that the number of younger patients and women of childbearing age undergoing bariatric surgery continues to increase and BMD measurement is not a routine follow-up investigation,[16] there is an urgent need to assess the impact of RYGB and SG on bone health in the UK bariatric population.

Adherence to a postbariatric lifestyle changes is the cornerstone of a successful weight loss.[17] Studies have shown that greater PA, lower sedentary time and high compliance to dietary recommendation postsurgery associate with greater weight loss and FM loss, preservation of lean muscle mass (LMM) and bone mass, as well as improvement in HRQoL.[18–22] However, patients spend 80% of their waking time in sedentary behaviour postsurgery,[23] activity that associates with increased risk of cardiometabolic disease and mortality.[24] Following surgery, patients are advised to undertake at least 150 min of moderate-to-vigorous physical activity (MVPA) per week, a duration and intensity that are recommended to reap the metabolic benefit of PA.[25] However, objectively measured MVPA decreases postsurgery with only 10% of patients achieving the recommended MVPA levels.[26] Likewise, a recent study undertaken in the UK has reported that weight loss postsurgery did not

correspond to improvement in MVPA and sedentary behaviour. However, the small sample size of this study (n=22) together with relatively short follow-up period limited its generalisability.[27] Further studies are therefore required to expand the information in this regard. In terms of dietary recommendations, daily protein intake of 60 g or more postsurgery is crucial for increasing satiety, preserving LMM, improving body composition and preventing against weight regain.[28–31] However, most patients are unable to achieve this in the first postoperative year, the period when rapid weight loss occurs.[32] Whether this is also the case for UK bariatric population is not known as no such data has ever been reported thus far.[32]

Impaired HRQoL is common in obesity[33] and often one of the driving factors for seeking weight loss surgery.[34] HRQoL is defined as individuals' perception of well-being that refers to physical, psychological and social domains of health.[35] Most studies reported improvement in all HRQoL domains with greater scores observed in the first postoperative year although some studies showed that the improvement is limited to only the physical domain but not the mental health component of HRQoL.[36] Despite mounting evidence in the international literature reporting the beneficial impact of bariatric surgery on HRQoL, data from the UK bariatric population does not exist.[37] There is some evidence that bariatric surgery can reduce in cost savings that offset the initial costs of surgery, though little UK evidence for RYGB and SG.[38–40]

Taken together the lack of postoperative data coupled with recommendations from systematic reviews[26 32] provide a strong rational to undertake a prospective study to evaluate the impact of RYGB and SG on body composition particularly BMD, PA levels, sedentary behaviour, physical function and strength, dietary intake, HRQoL and costs in a UK bariatric population. Information gained from this study will provide valuable data to inform the implementation of future postsurgery lifestyle programmes with the aim of maximising the beneficial outcomes of bariatric surgery as highlighted by NICE.[1] This paper details the study design and outcomes of the BARI-LIFESTYLE observational study.

## OBJECTIVES

The overall objective of BARI-LIFESTYLE observational study is to evaluate the impact of RYGB and SG on changes in body weight, body composition, PA levels, sedentary behaviour, physical function and strength, dietary intake, HRQoL, remission of comorbidities, healthcare resource utilisation and costs in a cohort of 100 patients.

The specific objectives are to evaluate postsurgery changes in:

1. percentage weight loss (%WL) at 1 year postsurgery, relative to baseline presurgery weight;
2. body FM, assessed using dual-energy X-ray absorptiometry (DXA) scan and bioelectrical impedance

analysis (BIA), relative to presurgery at 12 months postsurgery;

3. LMM, assessed using DXA scan and BIA, relative to presurgery at 12 months postsurgery;

4. BMD, assessed using DXA scan and BIA, relative to presurgery at 12 months postsurgery;

5. PA levels (light, moderate, vigorous), percentage achieving 150 min of MVPA in a week and sedentary behaviour assessed using accelerometer at 3, 6 and 12 months postsurgery, relative to presurgery;

6. physical function and strength assessed using 6 min walk test (6MWT), sit-to-stand (STS) test and hand-grip test at 3, 6 and 12 months postsurgery, relative to presurgery;

7. dietary intake assessed using food diary at 3, 6 and 12 months postsurgery, relative to presurgery;

8. HRQoL assessed using EuroQol-5D-3L (EQ-5D-3L) and Impact of Weight on Quality of Life-Lite (IWQOL-Lite) at 3, 6 and 12 months postsurgery, relative to presurgery;

9. characteristics of attitude and symptoms of depression assessed using Beck Depression Inventory-II (BDI-II) at 3, 6 and 12 months postsurgery, relative to presurgery;

10. obesity-associated comorbidities (type 2 diabetes (T2D), dyslipidaemia, hypertension, obstructive sleep apnoea (OSA)) at 3, 6 and 12 months postsurgery, relative to presurgery;

11. healthcare resource utilisation and costs assessed using an adapted version of the Client Service Receipt Inventory (CSRI) at 3, 6 and 12 months postsurgery, relative to presurgery.

## METHODS AND ANALYSIS
### Study design and setting
BARI-LIFESTYLE observational study is a prospective, longitudinal cohort study within routine clinical care setting of patients undergoing bariatric surgery conducted in London, UK (figure 1). A total of 100 patients who are planned to undergo either primary RYGB or SG will be recruited over a 2-year period from 2018 to 2019, and will be followed for up to 12 months postsurgery. Recruitment will take place at the Bariatric Centre for Weight Management and Metabolic Surgery, University College London Hospitals (UCLH) (study site) and the Bariatric and Obesity Surgery Clinic at the Whittington Hospital that acts as a participant identification centre (PIC). Participants recruited from the Whittington Hospital will have their surgical procedure undertaken at the same centre, but all study procedures such as written informed consent, baseline assessment, postsurgery follow-up care and study assessments will be undertaken by the bariatric team at UCLH. In both centres, the decision for procedure selection is based on informed patient preference after standardised counselling including details, potential risks, and benefits of each procedure that adheres to the current international

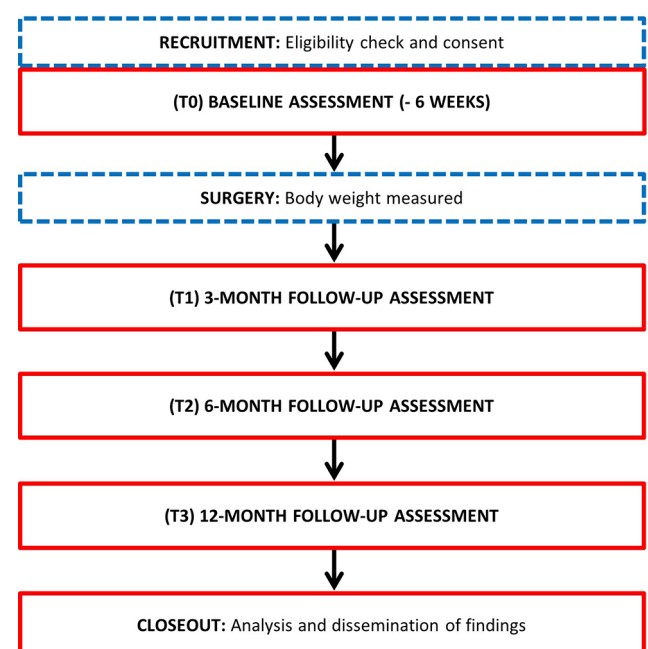

**Figure 1** Flow diagram of participant enrolment, consent, assessment and associated timeline.

guideline for the surgical recommendation for obesity and weight-related disease.[41] This study is carried out by the Centre for Obesity Research, Division of Medicine, University College London (UCL), with an expected total duration of 36 months, from the first participant enrolled to last participant follow-up.

### Participants and recruitment
Patients will be screened for suitability for the study by the bariatric team at the study site and PIC based on the inclusion and exclusion criteria when they attend the standard presurgical assessment (box 1). Verbal consent will be sought from those fulfilling the eligibility criteria and interested in participating to be approached by a research investigator. Patients will be given a participant information sheet inviting them to participate in a 1-year prospective, longitudinal cohort study looking at the effect of bariatric surgery on body weight, body composition, PA levels, sedentary behaviour, physical function and strength, dietary intake, HRQoL, remission of comorbidities, healthcare resource utilisation and costs. Consented participants will then be scheduled to attend a baseline assessment, approximately 6 weeks before surgery day at the study site. Each participant will be given a Fitbit Alta HR to enable them to self-monitor their activity levels and to reduce sedentary behaviour. Based on the weekly number of bariatric procedures undertaken at UCLH and Whittington Hospital and after considering the eligibility criteria, estimated recruitment rate is approximately seven participants per month. Hence, the expected recruitment period for the study is 15–20 months.

All participants will receive the standardised postbariatric care as stipulated by NICE.[1] Participants will

## Box 1 Participant eligibility criteria for participation in the BARI-LIFESTYLE observational study

### Inclusion criteria
► Adult aged between 18 and 65 years.
► Planned to undergo either primary RYGB or SG surgery and fulfilling NICE eligibility criteria for bariatric surgery.[1]
► Able to read and write in English.
► Willing and able to provide written informed consent.
► Able to comply with study protocol.
► Willing and able to wear a Fitbit wrist-based activity tracker device and an ActiGraph device.

### Exclusion criteria
► More than 200 kg of body weight due to the limitation of DXA scan.
► Non-ambulatory.
► Functional limitation.

DXA, dual-energy X-ray absorptiometry; NICE, National Institute for Health and Care Excellence; RYGB, Roux-en-Y gastric bypass; SG, sleeve gastrectomy.

attend the study site for monitoring of nutritional intake, vitamin and mineral deficiencies, comorbidities and medication review. Participants will receive verbal PA and dietary advice from a specialist nurse and dietitian at weeks 12 and 36 postsurgery, respectively.

### Outcomes measures
Outcome measures will be collected at four study time points, designed to coincide with normal follow-up care visits; baseline visit at approximately 6 weeks before surgery (T0) then at 3 (T1), 6 (T2) and 12 months (T3) postsurgery (table 1).

### Sociodemographic, medical history and physical examination
Participants' sociodemographic data, medical history and physical examination will be completed by the bariatric team at the baseline visit. Data to be captured including age, gender, ethnicity, educational level, marital status, medication intake, weight history, pregnancy history, alcohol consumption using the Alcohol Use Disorders Identification Test-C (AUDIT-C) questions,[42] smoking habits and family history of obesity and comorbidities.

### Primary outcome
Body weight will be measured using a weighing scale (Model VT200/220; Vishay Transducers, California, USA) with participants wearing light clothes and without shoes and heavy accessories, to the nearest 0.1 kg. Similarly, height will be determined using a stadiometer (Seca 242, Seca, Hamburg, Germany) to the nearest 0.01 m. %WL will be calculated using the following formula: %WL = ((weight on the day of surgery−weight at time point after

| Table 1 Study timeline and investigations | | | | | |
|---|---|---|---|---|---|
| | Baseline (T0) | Day of surgery | 3 months postsurgery (T1) | 6 months postsurgery (T2) | 12 months postsurgery (T3) |
| Sociodemographic data | ✓ | | | | |
| Height | ✓ | | | | |
| Weight | ✓ | ✓ | ✓ | ✓ | ✓ |
| Blood pressure and heart rate | ✓ | | ✓ | ✓ | ✓ |
| Dual-energy X-ray absorptiometry scan | ✓ | | | | ✓ |
| Bioelectrical impedance analysis | ✓ | | ✓ | ✓ | ✓ |
| Laboratory test | ✓ | | ✓ | ✓ | ✓ |
| Physical activity levels (ActiGraph) and activity diary | ✓ | | ✓ | ✓ | ✓ |
| Physical function and strength | | | | | |
| 6 min walk test | ✓ | | ✓ | ✓ | ✓ |
| Sit-to-stand | ✓ | | ✓ | ✓ | ✓ |
| Handgrip test | ✓ | | ✓ | ✓ | ✓ |
| Dietary intake (3-day food diary) | ✓ | | ✓ | ✓ | ✓ |
| Completion of questionnaires | | | | | |
| EuroQol-5D-3L | ✓ | | ✓ | ✓ | ✓ |
| Impact of weight on Quality of Life-Lite | ✓ | | ✓ | ✓ | ✓ |
| Beck Depression Inventory-II | ✓ | | ✓ | ✓ | ✓ |
| Client Service Receipt Inventory | ✓ | | ✓ | ✓ | ✓ |
| Review of medication | ✓ | | ✓ | ✓ | ✓ |
| Review of comorbidities | ✓ | | ✓ | ✓ | ✓ |

surgery)/weight on the day of surgery)×100, measured at each study time point.

## Secondary outcomes

### Body composition (body FM, LMM and BMD)

Body composition will be assessed at baseline and 12 months postsurgery using DXA scan (Discovery A DXA system, software V.13.4.2; Hologic; Massachusetts, USA). DXA scan uses ionising radiation to measure different body compartments. This is the current reference standard for assessing body composition and a gold standard method to diagnose osteopenia and osteoporosis.[43] In addition, body composition will be measured using BIA (Tanita DC-430MAS; Tanita, Tokyo, Japan) at each study visit. This is a non-invasive, easy to perform and cheaper option to measure body composition that is based on the differences in electrical conductivity of FM and FFM tissues.[44]

### PA levels and sedentary behaviour

PA and time spent in light, moderate and vigorous activities, and sedentary behaviour will be measured objectively using ActiGraph wGT3X-BT (Pensacola, Florida, USA), an accelerometer-based activity monitor.[45] Participants will be instructed to wear the ActiGraph on their dominant hip for 1 week, from waking in the morning until going to bed at night, and to remove it only during water-based activities. Additionally, participants will be asked to keep an activity diary throughout the week, to assist interpretation of data from the device. Both the device and activity diary have to be returned to the investigators for data analysis (ActiLife software V.6.13.3; Pensacola, Florida, USA).

### Physical function and strength

Participants' functional capacity will be assessed using a 6MWT, a self-paced, submaximal assessment of functional capacity used to prescribe appropriate exercise.[46] Lower body functional capacity and strength will be assessed using the STS test.[47] Static muscle strength will be assessed using Jamar Hydraulic Hand Dynamometer (Patterson Medical; Illinois, USA).[48]

### Dietary intake

All participants will be required to keep a 3 days food diary (two working days and one weekend day) for 1 week at each study time point. This method has a higher agreement with the 9 days food dairy compared with the food frequency questionnaire[49] while reducing the burden to patients and thus promoting better compliance for documenting food intake. The completed food diary will be returned to the investigators together with the ActiGraph and activity diary by using a stamped addressed envelope provided to participants.

### Health-related quality of life

HRQoL will be assessed using EQ-5D-3L and IWQOL-Lite. The EQ-5D-3L descriptive system is a 5-item self-report questionnaire that assesses the following domains:

mobility, self-care, usual activities, pain/discomfort and anxiety/depression, and a visual analogue scale, which records self-rated health on a 0–100 scale.[50] EQ-5D-3L health states will be converted into utility values using a formula that attaches weights to each level in each dimension based on valuations by general population samples. We will use a value set for the UK population to calculate utility values at each time point for every participant.[51] The IWQOL-Lite is a 31-item, self-report, obesity and overweight-specific measure of HRQoL.[52] This tool consists of a total score and scores on each of five scales – physical function, self-esteem, sexual life, public distress, and work; higher scores indicate better HRQoL.

### Attitude and symptoms of depression

BDI-II is a 21-item self-report questionnaire that assesses mood over the past week.[53] Symptoms of depression are classified by the total score: minimal, mild, moderate, and severe symptoms.

### Obesity-associated comorbidities

Comorbidities (T2D, dyslipidaemia, hypertension, OSA) and medication review will be carried out at each study time point.

### Healthcare resource utilisation and costs

Resource use data will be collected using an adapted version of the CSRI,[54] including the costs of bariatric surgery plus presurgery visits, number of contacts with healthcare professionals, visits to specialist clinics, the emergency department, admissions to the hospital, primary care contacts, and medications. Resource use data will be converted into costs using published unit costs.[55–57] In addition, information regarding support from informal carers, employment status and time off work will be collected. Resource use data will be collected for the previous 6 months at the baseline visit and since participants' last study visit at each postsurgery study time point.

### Sample size

A sample size of 100 patients will be enough to model the primary outcome and the range of secondary outcomes with a reasonable level precision and with regard to the number of patients who are likely to be recruited within the study's time frame. Also, a sample size of 100 patients will be sufficient to estimate the %WL at 1 year postsurgery to within ±2.5% using a 95% CI. This calculation accounts for a possible drop-out rate of up to 25% and assumes a conservative estimate for SD of %WL of 10%. This sample size should also ensure that there are enough data points for linear mixed effects models to be fitted with parameter estimates that have a satisfactory level of precision and where the model fitting algorithm will converge.

### Statistical analysis

The demographic and medical history information collected at baseline shall be presented in a table.

Categorical variables shall be reported as raw numbers and percentages. Reports of continuous variables shall include mean, median, range and SD.

### Primary outcome analysis

The primary outcome is the %WL measured longitudinally at baseline and 12 months postsurgery. %WL will be analysed using a linear mixed effects model over the three postsurgery time points (3 months, 6 months and 12 months) after controlling for the baseline body weight measure and height. Model assumptions shall be checked and suitable transforms of the primary outcome variable considered if necessary. In addition, overall percentage change in weight since baseline shall be computed marginally at each of 3, 6 and 12 months and displayed graphically.

### Secondary outcomes analyses

Analyses of longitudinal secondary outcomes shall be performed using linear mixed effects regression models, with a normal distribution assumed for continuous outcomes (or a suitable transform of these outcomes). Model parameter estimates together with appropriate 95% CIs shall be reported. Categorical outcomes (eg, proportions of participants with comorbidities) shall be summarised in tabular form at each time point. Where appropriate (for example, for proportions), estimates and 95% CIs will be presented. To analyse costs, we will assume the costs measured at baseline for the preceding 6 months would persist during follow-up in the absence of surgery; we will then compare postsurgery costs with predicted costs that would have been incurred in the absence of surgery. To account for skewness of the cost data, we will use a generalised linear model with gamma family and log link.[58]

### Missing data

Bias due to missing data will be investigated by comparing the baseline characteristics of participants with and without missing values. Depending on the extent of missingness, the predictors of missing values will be identified. The primary outcome analysis will be adjusted for those predictors of missing values, which are related to missingness. Multiple imputation using chained equations shall be considered as part of a sensitivity analysis for missing data in the primary outcome model.

### Data storage and retention

All data will be handled in accordance with the UK Data Protection Act 1998. Physical data will be stored in a secure room with limited access to only members of the research team, whereas computers storing electronic data will be encrypted and password protected. Each participant will be given a unique study identification number and used on their records instead of their name. The master list linking participants' name and the study identification number will be kept in a secure location. This way, participants' personal identity and data collected in the study cannot be linked by anyone outside the study team. This study is registered with the UCL Data Protection (Reference: Z6364106/2017/04/43). At the end of the study, all essential documentation will be archived securely for a minimum of 20 years from the declaration of the end of study.

### Ethics and dissemination

Potential participants will be explained in detail regarding the aims, methods, anticipated advantages and disadvantages of participation in the study by Good Clinical Practice trained investigators prior to obtaining their written informed consent. Participants will be informed that their participation is on a voluntary basis, and they have the right to withdraw from the study at any time without affecting their present and future medical care. No research procedures will be undertaken prior to patients giving written informed consent. As a duty of care, all possible adverse events will be collected from the day participants consented for the study to monitor their safety.

The findings will be presented to stakeholder groups locally, nationally and internationally and published in peer-reviewed medical journals. The lay-person summary of the findings will be published on the Centre for Obesity Research, UCL website (http://www.ucl.ac.uk/obesity). The results will be fully anonymised, and none of the participants will be identified in any report or publication.

### ADVANTAGE AND LIMITATION

This observational study will address the need for more high-quality data that examine the outcomes of RYGB and SG derived from the UK bariatric population. It will involve a comprehensive assessment and data collection at four study time points in the first year of surgery enabling an indepth analysis of changes in body composition, PA levels, sedentary behaviour, physical function and strength, dietary intake, HRQoL and costs, relative to presurgery. Data collection will be carried out by using validated assessment methods and questionnaires. Another advantage of this study is the use of DXA scan, a reference standard to measure body composition.[43] Also, the use of accelerometer will generate high-quality data to measure objective PA levels and sedentary behaviour. Studies have shown that bariatric surgery patients tend to over-report their PA levels when assessed using the conventional PA questionnaires.[26]

This protocol for an observational study is not without limitations. First, the study design does not include a conventional intensive lifestyle intervention (non-surgical) as a comparator group. Second, patients will not be randomised for surgical procedure as this study does not aim to compare between RYGB and SG but aims to examine 'real-world' clinical outcomes where the patient/healthcare professional make an informed choice about which procedures is best. However, data that will be generated from this study will allow us to power a subsequent randomised study. Third, a potential sample

selection bias due to the exclusion of patients with functional limitation (eg, cognitive impairment, walking difficulties) and/or non-ambulatory and patients with more than 200 kg of body weight owing to the weight limit of the DXA scan. Finally, given resource limitations, only approximately 100 patients will be recruited in this 1-year observational cohort study. Nevertheless, this sample size is adequate to generate indepth insights into the various outcomes of RYGB and SG.

## CONCLUSION

BARI-LIFESTYLE observational study will produce a comprehensive data on the broad range of RYGB and SG outcomes derived from the UK bariatric population that is still scarce in the literature. The information gained from this study will inform future lifestyle programmes for postbariatric surgery patients.

**Author affiliations**
[1]Centre for Obesity Research, Division of Medicine, University College London, London, UK
[2]Bariatric Centre for Weight Management and Metabolic Surgery, University College London Hospitals, London, UK
[3]Biomedical Research Centre, National Institute of Health Research University College London Hospitals, London, UK
[4]Institute of Sport, Exercise and Health, London, UK
[5]Department of Statistical Science, University College London, London, UK
[6]Department of Applied Health Research, University College London, London, UK
[7]The Hatter Cardiovascular Institute, Institute of Cardiovascular Science, University College London, London, UK

**Acknowledgements** The authors wish to thank Professor Rumana Omar for her input with regard to the statistical analysis plan. We gratefully acknowledge our Patient and Public Involvement group for their contribution to the study design as to ensure participants' acceptability. We would also like to thank all members of the Steering Committee and our research team at the Centre for Obesity Research, UCL for their invaluable inputs in the study.

**Contributors** RLB and FCJ designed the overall study and drafted the manuscript. AC coordinated the study to ensure GCP compliance. HK, JD and AK planned the assessment for dietary intake and HRQoL. BP and NL planned the assessment for PA levels, sedentary behaviour and physical function and strength. AGO advised on the statistical analysis plan. SM contributed to the analysis plan for healthcare resource usage and costs. AP and KC planned the assessment for body composition and review of comorbidities. RLB is the grant holder and chief investigator for the study. All authors have contributed to the refinement of the study protocol and editing the manuscript. All authors have read and approved the final manuscript.

**Funding** This study is supported by National Institute for Health Research (NIHR), the Sir Jules Thorn Charitable Trust, the Rosetrees Trust, the Stoneygate Trust and Robert Luff Foundation.

**Disclaimer** The funders were not involved in decisions relating to the study design and data collection. They will not have any role in the study execution, analyses, interpretation of data or writing of the manuscript and decision to submit results.

**Competing interests** None declared.

**Patient consent** Not required.

**Ethics approval** London-Dulwich Research Ethics Committee (Reference: 17/LO/0950).

**Provenance and peer review** Not commissioned; externally peer reviewed.

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
