## [Reviewer comments · BMJ Open]

ARTICLE DETAILS

TITLE (PROVISIONAL)	Protocol for a one-year prospective, longitudinal cohort study of patients undergoing Roux-en-Y gastric bypass and sleeve gastrectomy: the BARI-LIFESTYLE observational study.
AUTHORS	Jassil, Friedrich; Camemolla, Alisia; Kingett, Helen; Paton, Bruce; O'Keeffe, Aidan; Doyle, Jacqueline; Morris, Stephen; Lewis, Neville; Kirk, Amy; Pucci, Andrea; Chaiyasoot, Kusuma; Batterham, Rachel

VERSION 1 – REVIEW

REVIEWER	Gema Frühbeck Clínica Univ. de Navarra, University of Navarra, Spain
REVIEW RETURNED	03-Dec-2017

GENERAL COMMENTS	GENERAL COMMENTS The topic is highly interesting and relevant. Given the continuing obesity epidemic and the increasing number of patients undergoing bariatric surgery to try to collect high-quality data is extremely welcome. It is also important to produce country-specific information to have the possibility to compare with other settings. The manuscript is clearly written making it easy to read. The study protocol is exhaustive and addresses collection of high-quality data that are not obtained on a regular basis. The manuscript will benefit from considering a couple of additional details just to round it off: 1) Introduction: while body weight, BMI and EWL are the most frequently used measures to analyse bariatric surgery outcomes, these measurements do not provide accurate information on body composition. In this context, it would be important and worthwhile mentioning that currently eligibility and success criteria also point precisely to the need of more functional pre- and post-operative patient assessment [Frühbeck G. Bariatric and metabolic surgery: a shift in eligibility and success criteria. Nat Rev Endocrinol. 2015 Aug;11(8):465-77.] like the one pursued by the authors in their study protocol. 2) There is already existing information on the usefulness of body composition [Cummings DE, Cohen RV. Beyond BMI: the need for new guidelines governing the use of bariatric and metabolic surgery. Lancet Diabetes Endocrinol 2014;2:175-81. // Gómez-Ambrosi J et al. Dissociation of body mass index, excess weight loss and body fat percentage trajectories after 3 years of gastric bypass: relationship
---

	with metabolic outcomes. Int J Obes (Lond). 2017 Sep;41(9):1379-1387.] as well as physical activity [Vatier C, Henegar C, Ciangura C, Poitou-Bernert C, Bouillot JL, Basdevant A, Oppert JM. Dynamic relations between sedentary behavior, physical activity, and body composition after bariatric surgery. Obes Surg. 2012 Aug;22(8):1251-6.] assessment following bariatric surgery. It would be worthwhile mentioning this in the Introduction to further back the rationale of the study protocol. 3) Methods & Analysis: it might be worthwhile specifically mentioning that the protocol adheres to current international guidelines [De Luca M et al. Indications for Surgery for Obesity and Weight-Related Diseases: Position Statements from the International Federation for the Surgery of Obesity and Metabolic Disorders (IFSO). Obes Surg. 2016 Aug;26(8):1659-96.]. 4) Methods & Analysis: it is not quite clear if there will be a control intensive lifestyle group of patients with similar obesity deciding not to undergo bariatric surgery. The same applies to potential randomization to RYGB or SG. Obviously, from a medical point of view the indication to perform one or the other of the surgical techniques should not be randomized, in my view (but the protocol needs to describe the process and clarify if there will be randomization or not). 5) Advantage & Limitations: the presence or lack (as applicable) of a control conventional (not surgical) intensive lifestyle group as well as the randomization issues may be included here too.
--	---

REVIEWER	Antonio E. Pontiroli Università degli Studi di Milano, Italy
REVIEW RETURNED	08-Dec-2017

GENERAL COMMENTS	RYGB and SLG are the two most common bariatric surgery performed in the UK that result in comparable weight loss and remission of obesity associated co-morbidities. Based on the assumption that there is a paucity of studies examining the impact of RYGB and SLG upon body composition, physical activity levels, sedentary behaviour, physical function and strength, dietary intake, health-related quality of life and costs, the authors propose a one-year prospective, longitudinal cohort study aiming to recruit 100 patients with severe obesity undergoing either primary RYGB or SLG from two bariatric centres in London, UK. Participants will be followed-up four times during the study period; pre-surgery baseline (T0) and at 3 (T1), 6 (T2), and 12-month (T3) post-surgery. In addition to the standard follow-up investigations, assessments including dual-energy X-ray absorptiometry scan, bioelectric impedance analysis, 6-minute walk test, sit-to-stand test, and handgrip test will be undertaken together with completion of questionnaires. Physical activity levels and sedentary behaviour will be assessed using accelerometer, and dietary intake will be recorded using a 3-day food diary. Outcome measures will include: body weight, body fat mass, lean muscle mass, bone mineral density, physical activity levels, sedentary behaviour, physical function and strength, dietary intake, health-related quality of life, remission of co-morbidities, healthcare resource utilisation and costs. Criticism.
--

	The informations coming from this protocol would be, for the largest part, confirmatory of previous studies comparing RYGB and SLG, with similar numbers of patients and similar or longer observation periods. The new data would be represented by body composition, physical activity levels, sedentary behaviour, physical function and strength, dietary intake, health-related quality of life, remission of co-morbidities, healthcare resource utilisation and costs. Because of the inconsistencies of previous studies comparing RYGB and SLG, a prospective study should be randomized to yield reliable and meaningful results in comparing RYGB and SLG. Another aspect to be clarified is the population size. Based on previous studies, the authors should calculate the expected differences, the population size, and the power of analysis: why 100 patients and not 80 or 120?
--	---

VERSION 1 – AUTHOR RESPONSE

Response to Reviewers

Re: Protocol for a one-year prospective, longitudinal cohort study of patients undergoing Roux-en-Y gastric bypass and sleeve gastrectomy: the BARI-LIFESTYLE observational study

COMMENTS FROM REVIEWER 1

Reviewer Name: Gema Frühbeck

Institution and Country: Clínica Univ. de Navarra, University of Navarra, Spain

We thank this reviewer for the extremely helpful comments regarding our manuscript. For clarity, we have responded (in bold) within the original text of the reviewers' comments and, we have accordingly altered our manuscript in line with their suggestions.

The topic is highly interesting and relevant. Given the continuing obesity epidemic and the increasing number of patients undergoing bariatric surgery to try to collect high-quality data is extremely welcome. It is also important to produce country-specific information to have the possibility to compare with other settings. The manuscript is clearly written making it easy to read. The study protocol is exhaustive and addresses collection of high-quality data that are not obtained on a regular basis.

RESPONSE: We thank the reviewer for these positive comments regarding our manuscript and study rationale.

The manuscript will benefit from considering a couple of additional details just to round it off:

1) Introduction: while body weight, BMI and EWL are the most frequently used measures to analyse bariatric surgery outcomes, these measurements do not provide accurate information on body composition. In this context, it would be important and worthwhile mentioning that currently eligibility and success criteria also point precisely to the need of more functional pre- and post-operative patient assessment [Frühbeck G. Bariatric and metabolic surgery: a shift in eligibility and success criteria. Nat Rev Endocrinol. 2015 Aug;11(8):465-77.] like the one pursued by the authors in their study protocol.

RESPONSE: We thank the reviewer for highlighting this important point. We agree that the eligibility criteria and the parameters used to define successful outcomes of bariatric surgery should be beyond

body weight, BMI, and EWL. We have now added a sentence regarding this in the introduction together with the reference suggested (Page 4, 1st paragraph).

2) There is already existing information on the usefulness of body composition [Cummings DE, Cohen RV. Beyond BMI: the need for new guidelines governing the use of bariatric and metabolic surgery. *Lancet Diabetes Endocrinol* 2014;2:175-81. // Gómez-Ambrosi J et al. Dissociation of body mass index, excess weight loss and body fat percentage trajectories after 3 years of gastric bypass: relationship with metabolic outcomes. *Int J Obes (Lond)*. 2017 Sep;41(9):1379-1387.] as well as physical activity [Vatier C, Henegar C, Ciangura C, Poitou-Bernert C, Bouillot JL, Basdevant A, Oppert JM. Dynamic relations between sedentary behavior, physical activity, and body composition after bariatric surgery. *Obes Surg*. 2012 Aug;22(8):1251-6.] assessment following bariatric surgery. It would be worthwhile mentioning this in the Introduction to further back the rationale of the study protocol.

RESPONSE: We thank the reviewer for these comments. Indeed, post-surgery fat mass loss is favourable whereas fat free mass loss is detrimental hence the importance of assessing body composition in bariatric surgery patients prior to and following surgery. Undoubtedly, one of the strategies to preserve fat free mass is through increasing physical activity and reducing sedentary time. As suggested by the reviewer, we have now added a sentence regarding this to the introduction with the suggested reference (Page 4, 2nd paragraph) and cited the study by Vatier et al. (Page 5, 3rd paragraph, 2nd sentence).

3) Methods & Analysis: it might be worthwhile specifically mentioning that the protocol adheres to current international guidelines [De Luca M et al. Indications for Surgery for Obesity and Weight-Related Diseases: Position Statements from the International Federation for the Surgery of Obesity and Metabolic Disorders (IFSO). *Obes Surg*. 2016 Aug;26(8):1659-96.].

RESPONSE: As suggested by the reviewer, we have now added this important point in the Method & Analysis under the study design and setting (Page 8).

4) Methods & Analysis: it is not quite clear if there will be a control intensive lifestyle group of patients with similar obesity deciding not to undergo bariatric surgery. The same applies to potential randomization to RYGB or SG. Obviously, from a medical point of view the indication to perform one or the other of the surgical techniques should not be randomized, in my view (but the protocol needs to describe the process and clarify if there will be randomization or not).

RESPONSE: We apologise for not explaining this adequately and we have now expanded upon this in the Methods and Analysis section (Page 8). This observational study will only be undertaken in patients undergoing bariatric surgery and will not include a conventional intensive lifestyle intervention (non-surgical) as a comparator group. Furthermore, patients will not be randomised to the surgical technique. In both centres, the decision for procedure selection is based on informed patient preference after standardised counselling including details, potential risks, and benefits of each procedure. We have also acknowledged this as a weakness of our study design and have now emphasised this in the Advantage and Limitation section (Page 15).

5) Advantage & Limitations: the presence or lack (as applicable) of a control conventional (not surgical) intensive lifestyle group as well as the randomization issues may be included here too.

RESPONSE: We thank the reviewer for these suggestions. We have now included these points in the Advantage & Limitation section (Page 15).

COMMENTS FROM REVIEWER 2

Reviewer Name: Antonio E. Pontiroli

Institution and Country: Università degli Studi di Milano, Italy

We thank this reviewer for the extremely helpful comments regarding our manuscript. For clarity, we have responded (in bold) within the reviewer's original text of the comments and, we have accordingly altered our manuscript in line with their suggestions.

RYGB and SLG are the two most common bariatric surgery performed in the UK that result in comparable weight loss and remission of obesity associated co-morbidities. Based on the assumption that there is a paucity of studies examining the impact of RYGB and SLG upon body composition, physical activity levels, sedentary behaviour, physical function and strength, dietary intake, health-related quality of life and costs, the authors propose a one-year prospective, longitudinal cohort study aiming to recruit 100 patients with severe obesity undergoing either primary RYGB or SLG from two bariatric centres in London, UK. Participants will be followed-up four times during the study period; pre-surgery baseline (T0) and at 3 (T1), 6 (T2), and 12-month (T3) post-surgery. In addition to the standard follow-up investigations, assessments including dual-energy X-ray absorptiometry scan, bioelectric impedance analysis, 6-minute walk test, sit-to-stand test, and handgrip test will be undertaken together with completion of questionnaires. Physical activity levels and sedentary behaviour will be assessed using accelerometer, and dietary intake will be recorded using a 3-day food diary. Outcome measures will include: body weight, body fat mass, lean muscle mass, bone mineral density, physical activity levels, sedentary behaviour, physical function and strength, dietary intake, health-related quality of life, remission of co-morbidities, healthcare resource utilisation and costs.

Criticism.

The information coming from this protocol would be, for the largest part, confirmatory of previous studies comparing RYGB and SLG, with similar numbers of patients and similar or longer observation periods. The new data would be represented by body composition, physical activity levels, sedentary behaviour, physical function and strength, dietary intake, health-related quality of life, remission of co-morbidities, healthcare resource utilisation and costs.

Because of the inconsistencies of previous studies comparing RYGB and SLG, a prospective study should be randomized to yield reliable and meaningful results in comparing RYGB and SLG.

RESPONSE: We thank the reviewer for raising this point and we apologise for not adequately explaining that the aim of our current study is to investigate "real world (albeit real UK)" changes that occur in patients currently undergoing RYGB and SG in terms of impact upon body composition, physical activity levels, sedentary behaviour, physical function and strength, dietary intake, health-related quality of life, remission of co-morbidities, healthcare resource utilisation and costs (additional information added in the Abstract at page 2 and Method and Analysis section at page 7). We do not aim to compare between RYGB and SG with our current study. However, our study will provide data in order to allow us to power a subsequent randomised study. We acknowledge that an appropriately powered randomised study is warranted and we have added a sentence regarding this as a weakness of our study design and have now emphasised in the Advantage and Limitation section (Page 15). In both centres, the decision for procedure selection is based on informed patient

preference after standardised counselling including details, potential risks, and benefits of each procedure. We have now detailed this in the Method and Analysis section (Page 8).

Another aspect to be clarified is the population size. Based on previous studies, the authors should calculate the expected differences, the population size, and the power of analysis: why 100 patients and not 80 or 120?

RESPONSE: We thank the reviewer for raising this point and we apologise for not detailing the sample size justification. As there is no comparator group, the calculation of expected differences would not typically be undertaken. In addition, this would make a traditional sample size calculation based on an expected difference between groups inappropriate for this observational study.

The sample size of 100 patients was used so that there would be enough data to model the primary outcome and the range of secondary outcomes with a reasonable level precision and with regard to the number of patients who are likely to be recruited within the study's time frame. A sample size of 100 patients will be sufficient to estimate the percentage weight loss at one year post-surgery to within $\pm 2.5\%$ using a 95% confidence interval. This calculation accounts for a possible drop-out rate of up to 25% and assumes a conservative estimate for standard deviation of percentage weight loss of 10%.

This sample size should also ensure that there are enough data points for linear mixed effects models to be fitted with parameter estimates that have a satisfactory level of precision and where the model fitting algorithm will converge. We have now added these justifications in the manuscript (Page 12, Sample size).

VERSION 2 – REVIEW

REVIEWER	Gema Frühbeck Clínica Univ. de Navarra, University of Navarra, Pamplona, Spain
REVIEW RETURNED	08-Jan-2018

GENERAL COMMENTS	The authors have satisfactorily addressed all the issues raised.
--

REVIEWER	Antonio Pontiroli Università degli Studi di Milano
REVIEW RETURNED	29-Jan-2018

GENERAL COMMENTS	I have read with interest the replies of authors to the two reviewers. I think that replies are adequate, and remove possible previous criticisms. there are no further comments or criticisms. the protocol is now valid and can be accepted. AE Pontiroli
--